# One-for-All: Generalized LoRA for Parameter-Efficient Fine-tuning

## Abstract

We present Generalized LoRA (GLoRA), a flexible approach for universal parameter-efficient fine-tuning tasks. Enhancing Low-Rank Adaptation (LoRA), GLoRA employs a generalized prompt module to optimize pre-trained model weights and adjust intermediate activations, providing more flexibility and capability across diverse tasks and datasets. Moreover, GLoRA facilitates efficient parameter adaptation by employing a scalable, modular, layer-wise structure search that learns individual adapter of each layer. Originating from a unified mathematical formulation, GLoRA exhibits strong transfer learning, few-shot learning and domain generalization abilities, as it adapts to new tasks through not only weights but also additional dimensions like activations. Comprehensive experiments demonstrate that GLoRA outperforms all previous methods in natural, specialized, and structured benchmarks in the vision field, achieving superior accuracy with fewer parameters and computations. To demonstrate the applicability in the language domain, we perform GLoRA on LLaMA-1 and 2, which also achieve considerable enhancements compared to the original LoRA. Furthermore, our structural re-parameterization design ensures that GLoRA incurs no extra inference cost, rendering it a practical solution for resource-limited applications.

## 1 Introduction

The remarkable achievements of large-scale deep neural networks in recent years have revolutionized the field of artificial intelligence, demonstrating unprecedented performance across various tasks and domains. These highly complex models, often with millions or even billions of parameters, have demonstrated remarkable capabilities in areas such as computer vision (Dosovitskiy et al., 2021), natural language understanding (Vaswani et al., 2017), and speech recognition (Radford et al., 2022). Typically, these colossal models are pre-trained on general and large-scale datasets, such as ImageNet (Deng et al., 2009a), and are subsequently adapted to downstream target scenarios through fine-tuning or transfer learning. Given the immense computational resources required by large pre-trained architectures, many parameter-efficient fine-tuning (PEFT) methods (Hu et al., 2021; Shen et al., 2021; Jia et al., 2022; Zhang et al., 2022; Luo et al., 2023) have been proposed. For instance, Low-Rank Adaptation (LoRA) (Hu et al., 2021) aims to reduce the number of trainable parameters by exclusively learning pairs of rank-decomposition matrices whilst keeping the original model parameter static. Adapter (Houlsby et al., 2019) implements bottleneck adapter modules and incorporates a modest number of task-specific parameters into a fixed pre-trained model. Similarly, Visual Prompt Tuning (VPT) (Jia et al., 2022) introduces a minimal number of learnable parameters to the input of the Transformer, leaving the entire backbone frozen during fine-tuning.

However, distinct downstream datasets often possess unique characteristics, such as natural, specialized, and structured data, which differ significantly in distribution and composition. A static fine-tuning strategy may not sufficiently account for these disparities, thereby hindering its capacity to adapt to diverse datasets. To rectify this, we propose a flexible, parameter-efficient fine-tuning scheme in this work to manage the variations of multiple downstream datasets within a consolidated formulation. Our approach presents a generalized version of LoRA from a unified parameter-efficient fine-tuning perspective, amplifying LoRA's capability, scalability, and adaptability by rescaling and shifting intermediate activations, in conjunction with implementing a structural re-parameterization design, etc. It is challenging to devise a unified method that integrates all adjustable dimensions and possibilities when tuning a pre-trained network, especially in the case of

transformer architectures, while our proposed approach presents a practicable solution to navigate this complexity.

In detail, our approach proposes a unified framework that can achieve comprehensive fine-tuning paradigms from a single formulation, i.e., a *One-for-All* fine-tuning architecture. It comprises a supernet, which, when optimized cost-effectively through evolutionary search, yields results that surpass those of prevailing fine-tuning methodologies necessitating expensive data-dependent hyperparameter search. The proposed approach exhibits the following advantages: (1) It concurrently takes into account multiple dimensions to enhance capability and flexibility during fine-tuning, encompassing weights, features, and input tokens. (2) It conducts an implicit search devoid of any manual hyperparameter tuning, thus justifying the increased training time. (3) It incurs no additional inference cost thanks to our structural re-parameterization architecture, whereby the extra fine-tuning parameters will be fused to the proximate projection weights post-training.

We conduct comprehensive experiments on VTAB-1K (Zhai et al., 2020), ImageNet (Deng et al., 2009a) and its variants (Recht et al., 2019; Wang et al., 2019; Hendrycks et al., 2021b;a), and Huggingface leaderboard benchmarks (Edward Beeching, 2023) for evaluating on language domain. The VTAB-1K dataset comprises 19 heterogeneous vision datasets, enveloping a broad spectrum of visual domains that include natural objects and scenes, textures and shapes, satellite imagery, among others. GLoRA surpasses all previous state-of-the-art PEFT methods by a substantial margin in terms of average accuracy. Additionally, we evaluate the model's few-shot learning capacity on five fine-grained visual recognition datasets, akin to prior works (Zhang et al., 2022; Jia et al., 2022), along with its ability for domain generalization and robustness on ImageNet-V2 (Recht et al., 2019), ImageNet-Sketch (Wang et al., 2019), ImageNet-A (Hendrycks et al., 2021b), and ImageNet-R (Hendrycks et al., 2021a) datasets. GLoRA significantly outperforms previous methods across all these benchmarks, without incurring any extra computational overhead during the inference phase.

Our contributions:

- We propose Generalized LoRA (GLoRA), a novel parameter-efficient fine-tuning framework. GLoRA enhances the low-rank adaptation approach with a more generalized prompt module design per layer, offering enhanced capability and flexibility in finetuning.

- GLoRA offers a unified framework that achieves universal fine-tuning paradigms from a single formulation, i.e., a *One-for-All* [1] fine-tuning architecture. During inference, the adapters yielded through GLoRA seamlessly integrate into the base network, resulting in no additional model weights. Thus, it incurs no extra inference computational load.

- We conduct extensive experiments on large vision (ViT-B) and language models (LLaMA-1 and 2) with downstream fine-tuning, few-shot learning, and domain generalization using various datasets. Our experimental results demonstrate that GLoRA outperforms all previous methods on these benchmarks while requiring only a small number of extra tunable parameters in training and no additional inference cost.

## 2 GLoRA

In this section, we start from providing a mathematical overview of existing state-of-the-art PEFT methods and discuss the advantages and disadvantages for them. Then, we introduce a unified formulation of integrating all existing SOTA PEFT methods and elaborate our proposed generalized LoRA in detail following this unified formulation perspective. After that, a structural re-parameterization design is presented to show the inference efficiency without additional cost. An evolutionary search for optimal layer-wise configurations is also introduced to achieve the goal of generalized LoRA. We further give the theoretical analysis and discussions on the higher capability of the proposed method.

### 2.1 PREVIOUS SOLUTIONS WITH LIMITATIONS

**Visual Prompt Tuning** (Jia et al., 2022): VPT introduces a small amount of task-specific learnable parameters into the input space while freezing the entire pre-trained Transformer backbone during

---

[1] *One-for-All* represents that one formulation can be transformed into various shapes of PEFT paradigms.

downstream fine-tuning. It proposes two strategies: VPT-Shallow, i.e., only input space has the trainable prompt:

$$\begin{aligned}[\mathbf{x}_1, \mathbf{Z}_1, \mathbf{E}_1] &= L_1\left([\mathbf{x}_0, \mathbf{P}, \mathbf{E}_0]\right) \\ [\mathbf{x}_i, \mathbf{Z}_i, \mathbf{E}_i] &= L_i\left([\mathbf{x}_{i-1}, \mathbf{Z}_{i-1}, \mathbf{E}_{i-1}]\right)\end{aligned} \tag{1}$$

where $P$ is a trainable prompt. $\mathbf{x}$ is the [CLS] token, $\mathbf{E}$ are the image patches. Prompts use $<1\%$ trainable parameters as compared to the original model.

VPT-Deep, i.e., every layer has the trainable prompt. The formulation is:

$$[\mathbf{x}_i, \dots, \mathbf{E}_i] = L_i\left([\mathbf{x}_{i-1}, \mathbf{P}_{i-1}, \mathbf{E}_{i-1}]\right) \tag{2}$$

VTP-Deep outperforms full fine-tuning on many vision tasks and also has better accuracy in a low data regime. However, VPT increases cost in the inference stage which is not negligible.

**AdaptFormer** (Chen et al., 2022): AdaptFormer introduces a parallel learnable branch of two linear layers and ReLU over the MLP block and updates only this path while freezing other parts.

$$\tilde{x}_\ell = \mathbf{ReLU}\left(\text{LN}\left(x'_\ell\right) \cdot \mathbf{W}_{\text{down}}\right) \cdot \mathbf{W}_{\text{up}} \tag{3}$$

$$x_\ell = \mathbf{MLP}\left(\text{LN}\left(x'_\ell\right)\right) + s \cdot \tilde{x}_\ell + x'_\ell \tag{4}$$

where $x'_\ell$ are the tokens after MHSA at the $\ell$-th layer. $\mathbf{W}_{\text{down}}$ and $\mathbf{W}_{\text{up}}$ are weights corresponding to a down-projection layer and an up-projection layer from the parallel branch, respectively. $s$ is a scale factor. AdaptFormer also increases the inference cost due to the presence of a parallel branch.

**LoRA** (Hu et al., 2021): LoRA proposes to freeze the pre-trained model weights and injects trainable low-rank decomposition matrices into each layer. It learns only the residual from pre-trained weight. Assuming $\mathbf{W}_0$, $\mathbf{b}_0$, $x$ are pre-trained weights, bias and input, let $f$ be a linear layer, thus $f(x) = \mathbf{W}_0 x + \mathbf{b}_0$. During fine-tuning, $\mathbf{W}_0$ and $\mathbf{b}_0$ are frozen, the learning process can be:

$$f(x) = \mathbf{W}_0 x + \Delta \mathbf{W} x + \mathbf{b}_0 = \mathbf{W}_{\text{LoRA}} x + \mathbf{b}_0 \tag{5}$$

where $\Delta \mathbf{W}$ is the low-rank decomposition weights that are learnable.

**Scaling & Shifting Features (SSF)** (Lian et al., 2022): SSF module scales and shifts features after every MLP, MHSA, Layernorm module during training, and performs re-parameterization during inference as it is a linear structure.

$$\boldsymbol{y} = \boldsymbol{\gamma} \odot x + \boldsymbol{\beta} \tag{6}$$

where $\boldsymbol{y}$ is the output features. $\boldsymbol{\gamma}$ and $\boldsymbol{\beta}$ are the scale and shift factors, $\odot$ is the dot product. This method has no increase in inference but the capability is limited to feature adaptation.

**FacT** (Jie & Deng, 2022): FacT proposes to use a tensorization-decomposition method to store the additional weight, the weights of the model are tensorized into a single 3D tensor, and their additions are then decomposed into lightweight factors. In fine-tuning, only the factors will be updated and stored.

$$f(x) = \mathbf{W}_0 x + \mathbf{b}_0 + \mathbf{U}\Sigma\mathbf{V} x = (\mathbf{W}_0 + \mathbf{U}\Sigma\mathbf{V}) x + \mathbf{b}_0 \tag{7}$$

where $\Delta \mathbf{W}$ in LoRA is decomposed into $\mathbf{U}$, $\mathbf{V}$ and $\Sigma$. This is *Tensor-Train* in FacT.

$$f(x) = \mathbf{W}_0 x + \mathbf{b}_0 + \mathbf{UCPV} x = (\mathbf{W}_0 + \mathbf{UCPV}) x + \mathbf{b}_0 \tag{8}$$

where $\Delta \mathbf{W}$ in LoRA is decomposed into $\mathbf{U}$, $\mathbf{C}$, $\mathbf{P}$ and $\mathbf{V}$. This is *Tucker* in FacT.

**RepAdapter** (Luo et al., 2023): RepAdapter inserts lightweight networks into the pre-trained models, and the additional parameters will be re-parameterized to the nearby projection weights after training. Adding sequential (not parallel) adapter to both MHA and MLP, adapter is linear thus can be re-parameterized, and has two layers: downsampling dense FC layer to downsample inputs; upsampling downsampled features that are divided into group, and each group has an upsampling layer. The group of upsampling layers can be merged into a single sparse upsampling layer and can be re-parameterized directly into the original MLP/MHSA. The formulation can be:

$$\begin{aligned} f(x) &= \mathbf{W}_0\left(x + \mathbf{W}_u\left(\mathbf{W}_d x + \mathbf{b}_d\right) + \mathbf{b}_u\right) + \mathbf{b}_0 \\ &= \left(\mathbf{W}_0 + \mathbf{W}_0\mathbf{W}_u\mathbf{W}_d\right) x + \mathbf{W}_0\mathbf{W}_u\mathbf{b}_d + \mathbf{W}_0\mathbf{b}_u + \mathbf{b}_0 \end{aligned} \tag{9}$$

where $\mathbf{W}_u$, $\mathbf{W}_d$, $\mathbf{b}_u$ and $\mathbf{b}_b$ are learnable weights and biases, respectively.

**Limitations:** In general, many existing PEFT methods such as (VPT, Adapter) increase the inference time since the proposed structure cannot be re-parameterized. Direct prompt tuning is also

hard to design as it brings in computational burden and requires hyper-parameter tuning i.e., how and where to place prompts. LoRA can be re-parameterized at inference but it doesn't scale up for larger matrices and the adaptation ability is constrained on weight space. SSF / Repadaptor cannot learn the weight change i.e., $\Delta\mathbf{W}$ in weight space, whereas LoRA / FacT cannot efficiently learn the scaling and shifting of feature change i.e., $\Delta\mathbf{H}$ in features space. Both feature and weight space need flexibility while performing transfer learning from a large model. Our proposed idea in this work attempts at: $\Delta\mathbf{W}$ tuning, $\Delta\mathbf{H}$ tuning, along with $\mathbf{W}$ and $\mathbf{H}$ scale and shift learning.

## 2.2 A Unified Formulation of One-for-All

For model fine-tuning, we propose a unified formulation that encompasses tuning in both weight and feature space along with VPT-Deep level prompt design. Additionally, we adopt a re-parameterization strategy to incorporate auxiliary parameters into the adjacent projection weights during the inference stage. Broadly speaking, our method serves as a *superset* of all prior solutions, i.e., one-for-all mechanism. By setting different support tensors to zero, our GLoRA can be reduced to any of these predecessor methods. Unlike NOAH (Zhang et al., 2022), our architecture can be succinctly articulated as a unified mathematical equation. The

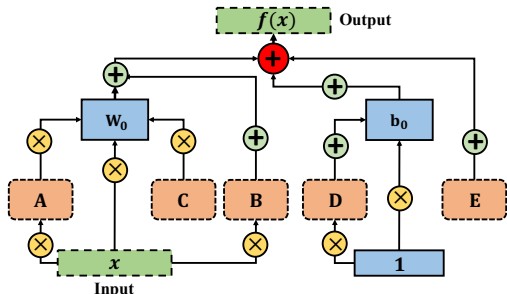

Figure 1: Schematic representation of a linear layer adapted with GLoRA.

consolidated formulation to represent all tunable spaces can be represented as follows:

$$f(x) = (\mathbf{W}_0 + \mathbf{W}_0\mathbf{A} + \mathbf{B})\,x + \mathbf{C}\mathbf{W}_0 + \mathbf{D}\mathbf{b}_0 + \mathbf{E} + \mathbf{b}_0 \tag{10}$$

where $\mathbf{A}$, $\mathbf{B}$, $\mathbf{C}$, $\mathbf{D}$, $\mathbf{E}$ are the trainable support tensors for downstream tasks in our GLoRA, $\mathbf{W}_0$ and $\mathbf{b}_0$ are frozen during whole fine-tuning. $\mathbf{A}$ is utilized to scale the weight. $\mathbf{B}$ has the role to scale the input and shift the weight. $\mathbf{C}$ is the layer-wise prompt serving a similar function of VPT-Deep, $\mathbf{D}$ and $\mathbf{E}$ are used to scale and shift the bias, respectively. A detailed illustration is shown in Figure 1.

**Module Design**. In this subsection, we delineate the methodology for designing layer-wise adaptors or prompt modules for $\mathbf{A}$, $\mathbf{B}$, $\mathbf{C}$, $\mathbf{D}$, $\mathbf{E}$. In a broad sense, these can take the form of `scalars`, `vectors`, `low-rank decompositions`, or `none`. Based on the role of these trainable support tensors, they can be sampled from the following respective search spaces:

$$\begin{aligned}
\mathbf{A} &= \{\text{LoRA}, \text{vector}, \text{scalar}, \text{none}\} \\
\mathbf{B} &= \{\text{LoRA}, \text{vector}, \text{scalar}, \text{none}\} \\
\mathbf{C} &= \{\text{LoRA}, \text{vector}, \text{none}\} \\
\mathbf{D} &= \{\text{vector}, \text{scalar}, \text{none}\} \\
\mathbf{E} &= \{\text{vector}, \text{scalar}, \text{none}\}
\end{aligned} \tag{11}$$

where `none` indicates zero, if all the trainable support tensors are zero, the model will be degraded to the original formulation and training recipe. In particular, suppose $\mathbf{W}_0 \in \mathbb{R}^{d\times d}$ is the original weight matrix. For every layer, we define $\mathbf{A}_d \in \mathbb{R}^{d\times r}$, $\mathbf{A}_u \in \mathbb{R}^{r\times d}$, $\mathbf{B}_d \in \mathbb{R}^{d\times r}$, $\mathbf{B}_u \in \mathbb{R}^{r\times d}$, $\mathbf{C}_d \in \mathbb{R}^{d\times r}$, $\mathbf{C}_u \in \mathbb{R}^{r\times 1}$, $\mathbf{D} \in \mathbb{R}^{d\times 1}$ and $\mathbf{E} \in \mathbb{R}^{d\times 1}$. We also define a multi-path supernet of all possible subnets and randomly sample a subnet during any given supernet training iteration for optimization. A subnet comprises of a single path network with different layerwise support tensors sampled from 11. Depending upon the current subnet configuration, in case of LoRA with rank $r_1 < r$, $\mathbf{A}_d^{r_1} \in \mathbb{R}^{d\times r_1}$, $\mathbf{A}_u^{r_1} \in \mathbb{R}^{r_1\times d}$ is indexed from $\mathbf{A}_d$ and $\mathbf{A}_u$ respectively; and $\mathbf{A} = \mathbf{A}_d^{r_1} \times \mathbf{A}_u^{r_1}$ is used as the final tensor, in case of vector $\mathbf{A} \in \mathbb{R}^{d\times 1}$ is indexed from $\mathbf{A}_d$ and in case of scalar $\mathbf{A} \in \mathbb{R}^{1\times 1}$ is indexed from $\mathbf{A}_d$. A similar strategy is followed for all other support tensors depending upon the current sampled configuration in the subnet. This weight entanglement strategy helps to increase the search space without increasing the number of parameters substantially and also shows faster convergence due to weight sharing in different subnets.

Moreover, without defining any existing adapter/module in the network explicitly, GLoRA proposes a much more *general formulation* that implicitly mimics the behavior of many existing works. In Table 1, we show how GLoRA is able to approximately mimic the behavior of many existing works by setting support tensors to specific attributes of the search space.

| Method | A | B | C | D | E |
|--------|------|------|--------|--------|------|
| LoRA | LoRA | None | None | None | None |
| VPT | None | None | Vector | None | None |
| SSF | Vector | None | Vector | Vector | None |
| RepAdapter | LoRA | None | None | Vector | None |

Table 1: Support tensor attributes for mimicking prior methods using GLoRA's formulation.

## 2.3 STRUCTURAL RE-PARAMETERIZATION DESIGN AND INFERENCE EFFICIENCY ANALYSIS

The fundamental aspect enabling re-parameterization (Ding et al., 2021) is the elimination of non-linearity amidst adjacent transformations, thereby permitting the absorption of supplementary parameters into the preceding ones. As mentioned in RepAdapter (Luo et al., 2023), the removal of such non-linear layers does not detrimentally impact the performance of the networks. The precise concept of GLoRA re-parameterization is explicated as follows:

$$f(x) = \mathbf{W}_{\text{uni}}x + \mathbf{b}_{\text{uni}} \tag{12}$$

where $\mathbf{W}_{\text{uni}}$ and $\mathbf{b}_{\text{uni}}$ are our final unified trained weight and bias in GLoRA. They are re-parameterized according to Eq 10:

$$\mathbf{W}_{\text{uni}} = \mathbf{W}_0 + \mathbf{W}_0\mathbf{A} + \mathbf{B} \tag{13}$$

$$\mathbf{b}_{\text{uni}} = \mathbf{C}\mathbf{W}_0 + \mathbf{D}\mathbf{b}_0 + \mathbf{E} + \mathbf{b}_0 \tag{14}$$

As a result, the re-parameterization strategy we employ, which integrates learnable parameters into the existing weight matrix offers a distinct advantage as it imposes no additional computational burden during the inference phase. This is further discussed in Section 4 where we provide thorough inference efficiency analysis of GLoRA compared to exisitng works.

## 2.4 EVOLUTIONARY SEARCH FOR OPTIMAL LAYER-WISE CONFIGURATIONS

Our design for a unified adaptor is implemented on a per-layer basis, thus allowing for heterogeneity across different layers. To identify the optimal configuration for each layer, we employ the evolutionary search method (Zhang et al., 2022; Shen et al., 2021), which offers a balance of efficiency and effectiveness. Although the training time may increase due to this search process, it is important to note that existing work (Zhang et al., 2022) necessitate an extensive hyperparameter search (such as low-rank in LoRA and FacT, as well as position and size of adapter modules in Adapter (Houlsby et al., 2019), dimension and structure configuration in RepAdapter (Luo et al., 2023), among others), as presented in the appendix. Our unified support tensor design conducts an implicit search that eliminates the need for manual hyperparameter tuning. Therefore, any augmentation in training time is reasonable and well-justified. More details regarding evolutionary search are in appendix. In the next section, we will discuss and explain the better capacity of our proposed GLoRA approach comparing to other counterparts for parameter-efficient fine-tuning task.

## 2.5 GLORA WITH HIGHER CAPACITY

Model capacity refers to the capability of a model to approximate a diverse range of functions. A method for regulating the capacity of a learning algorithm involves selecting an appropriate hypothesis space, essentially a set of functions that the learning algorithm is permitted to consider as potential solutions. The Vapnik-Chervonenkis Dimension (VC Dimension) (Vapnik & Chervonenkis, 2015) is a measure of the capacity (complexity, expressiveness) of a set of functions that can be learned by a statistical classification algorithm. It is defined as the cardinality of the largest set of points that the algorithm can shatter. By estimating the VC Dimension of a deep model, we can get an idea of how capable the model is of fitting complex datasets. A very high VC Dimension could indicate that the model has enough capacity to learn the training data perfectly but might overfit and generalize poorly on new data.

**Definition of VC Dimension**. The VC Dimension of a hypothesis class $\mathcal{H}$ (a set of functions) is the largest number of points that can be shattered by $\mathcal{H}$. A set of points is said to be shattered by $\mathcal{H}$ if, for every possible labeling (binary classification) of these points, there exists a hypothesis in $\mathcal{H}$ that

perfectly classifies the points according to that labeling. Mathematically, if we have a set of points $S = \{x_1, x_2, \ldots, x_d\}$, the hypothesis class $\mathcal{H}$ shatters $S$ if:

$$\forall y \in \{0, 1\}^d, \exists h \in \mathcal{H} : \forall i \in \{1, 2, \ldots, d\}, h(x_i) = y_i \tag{15}$$

The VC Dimension, denoted as $\mathbf{d}_{\mathrm{vc}}(\mathcal{H})$, is the maximum size of any set $S$ that can be shattered by $\mathcal{H}$. If $\mathcal{H}$ can shatter a set of size $d$ but cannot shatter any set of size $d + 1$, then $\mathbf{d}_{\mathrm{vc}}(\mathcal{H}) = d$.

**Theorem 1** *Suppose* $\mathbf{d}_{\mathrm{vc}}(\mathcal{H})$ *is the VC dimension of any finite hypothesis* $\mathcal{H}$. *If* $\mathcal{H}_{\mathrm{i}} \subseteq \mathcal{H}_{\mathrm{uni}}$,

$$\mathbf{d}_{\mathrm{vc}}(\mathcal{H}_{\mathrm{uni}}) - \mathbf{d}_{\mathrm{vc}}(\mathcal{H}_{\mathrm{i}}) \geq \epsilon \quad s.t. \quad \epsilon \geq 0$$

In the context of GLoRA, $\mathcal{H}_{\mathrm{i}}$ denotes the hypothesis space of a randomly sampled subnet and $\mathcal{H}_{\mathrm{uni}}$ denotes the hypothesis space of the complete supernet. The validity of this theorem stems from the inherent property of our problem context, where the hypothesis space $\mathcal{H}_{\mathrm{i}}$ is a subset of $\mathcal{H}_{\mathrm{uni}}$ in our context. $\mathcal{H}_{\mathrm{uni}}$ encompasses all possible shattered scenarios of $\mathcal{H}_{\mathrm{i}}$. For the extreme case where the VC dimension $\mathbf{d}_{\mathrm{vc}}(\mathcal{H}_{\mathrm{o}})$ ($\mathcal{H}_{\mathrm{o}}$ is the difference set of $\mathcal{H}_{\mathrm{uni}}$ and $\mathcal{H}_{\mathrm{i}}$) is 0, the error $\epsilon$ will be zero. As per learning theory, a higher VC dimension implies greater model flexibility and capability of our approach. Clearly, Theorem 1 holds for GLoRA and thus it experiences a greater model capacity.

## 3 EXPERIMENTS

**Datasets.** We thoroughly evaluate GLoRA on VTAB-1K (Zhai et al., 2020) benchmark for various parameter budgets. VTAB-1K comprises 19 image classification tasks clustered into three domains: i) Natural images ii) Specialized tasks consisting of remote sensing and medical datasets; and iii) Structured tasks focusing on scene structure understanding. To test the ability on few-shot learning, we evaluate GLoRA on five fine-grained visual recognition few-shot datasets: Food101 (Bossard et al., 2014), OxfordFlowers102 (Nilsback & Zisserman, 2006), StandfordCars (Krause et al., 2013), OxfordPets (Parkhi et al., 2012), and FGVCAircraft (Maji et al., 2013). Following previous work (Jie & Deng, 2022), we evaluate 1, 2, 4, 8, and 16 shot settings. Next, to show the domain generalization capabilities of GLoRA, we train it on ImageNet (Deng et al., 2009b) for a 16-shot setting and test on four out-of-domain datasets including ImageNetV2 (Recht et al., 2019), ImageNet-Sketch (Wang et al., 2019), ImageNet-A (Hendrycks et al., 2021b), and ImageNet-R (Hendrycks et al., 2021a). Finally, we show the performance of GLoRA on the Open LLM Leaderboard which consists of four datasets with varying prompt shots, namely AI2 Reasoning Challenge (25-shot) (Clark et al., 2018), TruthfulQA (0-shot) (Lin et al., 2022), HellaSwag (10-shot) (Zellers et al., 2019) and MMLU (5-shot) (Hendrycks et al., 2020).

**Network Architecture and Implementation Details.** For all the vision experiments, we utilize ViT-B (Dosovitskiy et al., 2021), a model pre-trained on ImageNet-21K, as our foundational model. For the language experiments, we consider two foundational base models: LLaMA-1-7B (Touvron et al., 2023a) and LLaMA-2-7B (Touvron et al., 2023b).

Our supernets undergo a training process spanning 500 epochs and 15 epochs for vision and language datasets respectively, operating with a constant batch size of 64 and a cosine learning rate scheduler. It is crucial to highlight that this precise policy demonstrates robust efficacy across all settings, regardless of the dataset in use. Post the training of supernet, we randomly sample 50 subnets from the supernet and then perform an evolutionary search for 20 and 5 epochs on vision and language tasks, respectively. Each step of random pick / crossover / mutation produces 50 new subnets. The probability for crossover and mutation is set to 0.2. Note that we did not perform any hyperparameter search over the evolution hyperparameters, and hence the performance might even improve after tuning the evolution hyperparameters. Finally, we report the performance of the searched subnet on the test set. The appendix provides further insights into dataset-specific learning rates and specific settings for different datasets.

### 3.1 RESULTS ON VTAB-1K

We train three different GLoRA supernet configurations to vary the number of trainable parameters. The difference in these is only the LoRA dimensions in the search space which varies from 8 and 4 in the largest model, 4 and 2 in the intermediate model, and 2 in the smallest model. This added parameter flexibility in our method allows for user-defined trainable parameter count in the final

models. Results on the VTAB-1k benchmark are shown in Table 2. We push the state-of-the-art in parameter-efficient transfer learning by up to 2.9%, Even our smallest model outperforms all existing methods by a substantial margin. It is worth noting that GLoRA performs competitively across datasets in contrast to all existing works which fail on at least one dataset, proving GLoRA's high generalization capabilities. GLoRA pushes the state of the art in as many as 14 out of 19 datasets in the VTAB-1k benchmark while performing competitively on the remaining datasets too.

Table 2: **Full results on VTAB-1K benchmark**. "# params" specifies the number of trainable parameters in backbones. Average accuracy and # params are averaged over group-wise mean values.

| | # param (M) | Inference Cost | Natural | | | | | | | Specialized | | | | Structured | | | | | | | | Average |
|---|---|---|---|---|---|---|---|---|---|---|---|---|---|---|---|---|---|---|---|---|---|---|
| | | | Cifar100 | Caltech101 | DTD | Flower102 | Pets | SVHN | Sun397 | Camelyon | EuroSAT | Resisc45 | Retinopathy | Clevr-Count | Clevr-Dist | DMLab | KITTI-Dist | dSpr-Loc | dSpr-Ori | sNORB-Azim | sNORB-Ele | |
| *Traditional Finetuning* | | | | | | | | | | | | | | | | | | | | | | |
| Full | 85.8 | - | 68.9 | 87.7 | 64.3 | 97.2 | 86.9 | 87.4 | 38.8 | 79.7 | 95.7 | 84.2 | 73.9 | 56.3 | 58.6 | 41.7 | 65.5 | 57.5 | 46.7 | 25.7 | 29.1 | 68.9 |
| Linear | 0 | - | 64.4 | 85.0 | 63.2 | 97.0 | 86.3 | 36.6 | 51.0 | 78.5 | 87.5 | 68.5 | 74.0 | 34.3 | 30.6 | 33.2 | 55.4 | 12.5 | 20.0 | 9.6 | 19.2 | 57.6 |
| *PEFT methods* | | | | | | | | | | | | | | | | | | | | | | |
| BitFit | 0.10 | - | 72.8 | 87.0 | 59.2 | 97.5 | 85.3 | 59.9 | 51.4 | 78.7 | 91.6 | 72.9 | 69.8 | 61.5 | 55.6 | 32.4 | 55.9 | 66.6 | 40.0 | 15.7 | 25.1 | 65.2 |
| VPT-Shallow | 0.06 | ↑ | 77.7 | 86.9 | 62.6 | 97.5 | 87.3 | 74.5 | 51.2 | 78.2 | 92.0 | 75.6 | 72.9 | 50.5 | 58.6 | 40.5 | 67.1 | 68.7 | 36.1 | 20.2 | 34.1 | 67.8 |
| VPT-Deep | 0.53 | ↑ | **78.8** | 90.8 | 65.8 | 98.0 | 88.3 | 78.1 | 49.6 | 81.8 | 96.1 | 83.4 | 68.4 | 68.5 | 60.0 | 46.5 | 72.8 | 73.6 | 47.9 | 32.9 | 37.8 | 72.0 |
| Adapter | 0.16 | ↑ | 69.2 | 90.1 | 68.0 | 98.8 | 89.9 | 82.8 | 54.3 | 84.0 | 94.9 | 81.9 | 75.5 | 80.9 | 65.3 | 48.6 | 78.3 | 74.8 | 48.5 | 29.9 | 41.6 | 73.9 |
| AdaptFormer | 0.16 | ↑ | 70.8 | 91.2 | 70.5 | 99.1 | 90.9 | 86.6 | 54.8 | 83.0 | 95.8 | 84.4 | **76.3** | 81.9 | 64.3 | 49.3 | 80.3 | 76.3 | 45.7 | 31.7 | 41.1 | 74.7 |
| LoRA | 0.29 | - | 67.1 | 91.4 | 69.4 | 98.8 | 90.4 | 85.3 | 54.0 | 84.9 | 95.3 | 84.4 | 73.6 | 82.9 | **69.2** | 49.8 | 78.5 | 75.7 | 47.1 | 31.0 | 44.0 | 74.5 |
| NOAH | 0.36 | ↑ | 69.6 | 92.7 | 70.2 | 99.1 | 90.4 | 86.1 | 53.7 | 84.4 | 95.4 | 83.9 | 75.8 | 82.8 | 68.9 | 49.9 | 81.7 | 81.8 | 48.3 | 32.8 | **44.2** | 75.5 |
| FacT | 0.07 | - | 70.6 | 90.6 | 70.8 | 99.1 | 90.7 | 88.6 | 54.1 | 84.8 | 96.2 | 84.5 | 75.7 | 82.6 | 68.2 | 49.8 | 80.7 | 80.8 | 47.4 | 33.2 | 43.0 | 75.6 |
| SSF | 0.24 | - | 69.0 | 92.6 | 75.1 | 99.4 | 91.8 | 90.2 | 52.9 | 87.4 | 95.9 | 87.4 | 75.5 | 75.9 | 62.3 | 53.3 | 80.6 | 77.3 | **54.9** | 29.5 | 37.9 | 75.7 |
| RepAdapter | 0.22 | - | 72.4 | 91.6 | 71.0 | 99.2 | 91.4 | 90.7 | 55.1 | 85.3 | 95.9 | 84.6 | 75.9 | 82.3 | 68.0 | 50.4 | 79.9 | 80.4 | 49.2 | 38.6 | 41.0 | 76.1 |
| **GLoRA** | 0.86 | - | 76.4 | **92.9** | 74.6 | 99.6 | **92.5** | **91.5** | **57.8** | 87.3 | **96.8** | 88.0 | 76.0 | **83.1** | 67.3 | **54.5** | **86.2** | **83.8** | 52.9 | **37.0** | 41.4 | **78.0** |
| **GLoRA** | 0.44 | - | 76.5 | 92.3 | 75.2 | 99.6 | 92.3 | 91.2 | 57.5 | 87.3 | 96.7 | 88.1 | 76.1 | 80.6 | 67.2 | 53.4 | 84.5 | 83.5 | 52.8 | 35.2 | 40.8 | 77.6 |
| **GLoRA** | 0.29 | - | 76.1 | 92.7 | **75.3** | **99.6** | 92.4 | 90.5 | 57.2 | **87.5** | 96.7 | **88.1** | 76.1 | 81.0 | 66.2 | 52.4 | 84.9 | 81.8 | 53.3 | 33.3 | 39.8 | 77.3 |

## 3.2 RESULTS ON LARGE LANGUAGE MODELS

Table 3: Performance of GLoRA on few-shot generative language tasks with LLM as backbones.

| Model | Dataset | Param (M) | ARC (25-s) | HellaSwag (10-s) | MMLU (5-s) | TruthfulQA (0-s) | Average |
|---|---|---|---|---|---|---|---|
| LLaMA-1-7B | - | - | 51.0 | 77.8 | 35.7 | 34.3 | 49.7 |
| LoRA | Alpaca | 3.1 | 53.5 | 77.3 | 33.8 | 34.8 | 49.8 |
| GLoRA | Alpaca | 1.9 | 52.9 | 78.1 | 34.5 | 37.8 | 50.8 |
| LoRA | ShareGPT | 3.1 | 51.7 | 77.9 | 36.1 | 39.2 | 51.2 |
| GLoRA | ShareGPT | 2.2 | 53.2 | 77.4 | 36.2 | 43.9 | 52.7 |
| LLaMA-2-7B | - | - | 53.1 | 78.5 | 46.9 | 38.8 | 54.3 |
| LoRA | ShareGPT | 3.1 | 53.3 | 78.4 | 45.8 | 41.2 | 54.7 |
| GLoRA | ShareGPT | 1.8 | 53.7 | 78.5 | 46.5 | 45.1 | 56.1 |

We apply GLoRA for LLMs by solely tuning the attention layers. This contrasts with vision tasks where all linear layers are adapted, to maintain a fair comparison with vanilla LoRA. We start from the publicly available LLaMA-1-7B (Touvron et al., 2023a) and LLaMA-2-7B (Touvron et al., 2023b) models and finetune it on the Alpaca (Taori et al., 2023) and ShareGPT dataset with only GLoRA support tensors trainable. For the evolutionary search, we use 5% random data sampled from the 4 given datasets for model validation during the evolutions. We finally report the searched model's performance on the standard Open LLM Leaderboard[2]. GLoRA consistently outperforms the pre-trained LLM and the corresponding LoRA fine-tuned variants. We keep the hyperparameters consistent between LoRA and GLoRA for a fair comparison, more details are in the appendix.

## 3.3 FEW-SHOT LEARNING

To extend the evaluation of GLoRA under conditions of limited data availability, we present the performance of GLoRA on fine-grained visual recognition datasets as the few-show learning, comparing it with LoRA, Adapter, VPT, and NOAH. The results at 1, 2, 4, 8, and 16 shots are illustrated in Figure 2 and Figure 6 of appendix. GLoRA demonstrates superior performance across the majority

---

[2]https://huggingface.co/spaces/HuggingFaceH4/open_llm_leaderboard

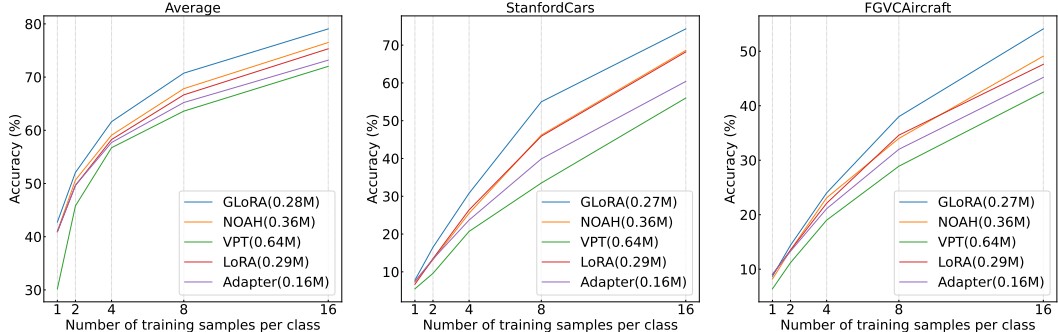

Figure 2: **Results on few-shot learning datasets**. The baseline methods include Adapter, LoRA, VPT, NOAH. GLoRA consistently performs better across five datasets and a varying number of training examples per class.

of the few-shot learning datasets, consistently outperforming the performance of existing methods by a large margin with similar parameter counts. Interestingly, on the Flowers102 dataset, all methods yield similar accuracy levels, attributable to the already exceptional overall performance. On the Food101 dataset, the average accuracy of GLoRA is on par with NOAH. From the first plot, we can observe, the average performance boost becomes more pronounced at higher shot scenarios, nevertheless, even at lower shot settings, the gains remain significant.

## 3.4 DOMAIN GENERALIZATION

Table 4: **Results on domain generalization**. GLoRA is significantly better than the existing works.

| | Source | Target | | | |
|---|---|---|---|---|---|
| | ImageNet | -Sketch | -V2 | -A | -R |
| Adapter Houlsby et al. (2019) | 70.5 | 16.4 | 59.1 | 5.5 | 22.1 |
| VPT Jia et al. (2022) | 70.5 | 18.3 | 58.0 | 4.6 | 23.2 |
| LoRA Hu et al. (2021) | 70.8 | 20.0 | 59.3 | 6.9 | 23.3 |
| NOAH Zhang et al. (2022) | 71.5 | 24.8 | 66.1 | 11.9 | 28.5 |
| GLoRA (0.29M) | 78.3 | 30.6 | 67.5 | 13.3 | 31.0 |

The capacity of out-of-domain generalization holds significant value for large-scale neural networks (Zhou et al., 2021). Models fine-tuned via PETL methods should exhibit enhanced domain generalization aptitude, thereby making them more applicable in real-world scenarios. We demonstrate the out-of-domain generalization capabilities of GLoRA in Table 4, where a single ImageNet-1K (Deng et al., 2009b) fine-tuned GLoRA model is subjected to testing on out-of-domain datasets. Aligning with preceding research, we limit the number of training examples per class to 16 for this experiment. It is noteworthy that the performance for the fully-scaled ImageNet-1K fine-tuned model stands at 83.97% (Dosovitskiy et al., 2021), and our approach manages to narrow this performance gap, even within a 16-shot setting (78.3%), thereby exhibiting superior few-shot learning on ImageNet-level datasets. Furthermore, the out-of-domain performance also witnesses a substantial boost in comparison to existing methods. When compared with LoRA, GLoRA enhances out-of-domain performance by as much as 100% (ImageNet-A) and 50% (ImageNet-Sketch).

## 4 ANALYSIS AND DISCUSSION

**Computational Cost.** We show the final inference throughput of various PEFT methods in Table 5, computed on an NVIDIA 3090 GPU. It is in this context that GLoRA significantly outperforms other methods, as GLoRA benefits from zero parameter or FLOPs overhead during the inference process. An ancillary advantage is the expedited adaptability in real-world scenarios where previous models are already deployed. The weights of GLoRA can be directly loaded without necessitating any manual system modifications. As previously mentioned, GLoRA supports VPT-Deep level prompts via the support tensor **C**, however, it does not impose any computational overhead due to its complete structural re-parameterization design.

**Visualizations of searched fine-tuning strategy for each layer.** Figure 4 visually shows the distribution of trainable parameters across the four types of linear layers embodied in ViT-B. Notably, the

Table 5: Inference efficiency comparison of GLoRA with existing methods.

| Method | ↑ #Param(M) | ↑ FLOPs(G) | Throughput (imgs/sec) | | |
|---|---|---|---|---|---|
| | | | bs = 1 | bs = 4 | bs = 16 |
| Full tuning | 0 | 0 | 91.5 | 375.7 | 539.5 |
| VPT Jia et al. (2022) | 0.55 | 5.60 | 86.1 | 283.5 | 381.5 |
| Adapter Houlsby et al. (2019) | 0.16 | 0.03 | 70.9 | 306.6 | 504.7 |
| AdaptFormer Chen et al. (2022) | 0.16 | 0.03 | 71.4 | 309.9 | 508.1 |
| NOAH Zhang et al. (2022) | 0.12 | 0.02 | 72.1 | 312.7 | 492.9 |
| LoRA Hu et al. (2021) GLoRA | 0 | 0 | 91.5 | 375.7 | 539.6 |

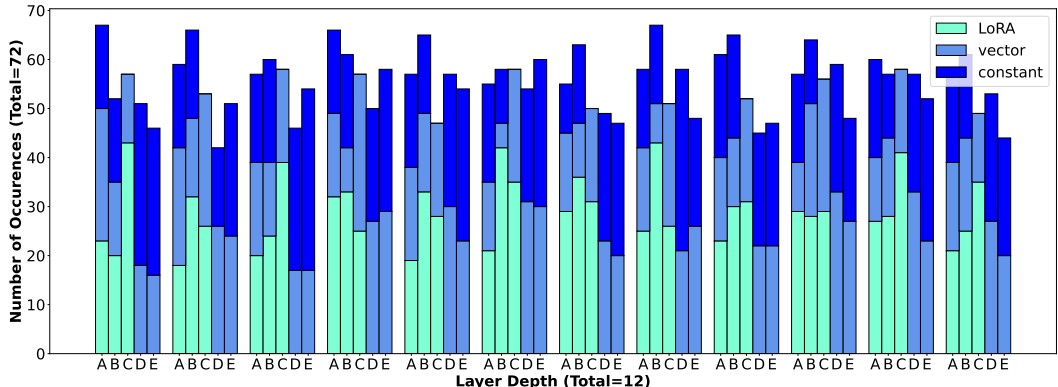

Figure 3: Layerwise configuration of support tensors of GLoRA (0.86M) on VTAB-1K dataset.

projection layer possesses the minimum quantity of trainable parameters spanning across VTAB-1K categories. More details of the searched fine-tuning strategy are discussed in Appendix.

## 5    RELATED WORK

Given the rapid expansion in model size, numerous methods for parameter-efficient fine-tuning (PEFT) have been introduced in the field of Natural Language Processing (NLP) to streamline the optimization of large language models (LLMs). (Liu et al., 2021a; Zhang et al.; Hu et al.; Liu et al., 2021b; Li & Liang, 2021; Lester et al., 2021; Zaken et al., 2022; Houlsby et al., 2019). The effectiveness of parameter-efficient fine-tuning has been proven in a wide range of natural language processing tasks (Fu et al., 2022). With the advent growth in the size of vision models (Dehghani et al., 2023; Kolesnikov et al., 2020), methods specifically focused on image models have also been put forward (Jie & Deng, 2022; Lian et al., 2022; Chen et al., 2022; Luo et al., 2023; Zhang et al., 2022; Jia et al., 2022; He et al., 2023). LoRA (Hu et al.) has proven to be effective across modalities.

## 6    CONCLUSION

We have presented GLoRA, a generalized parameter-efficient fine-tuning approach that has successfully demonstrated the effectiveness in enhancing the fine-tuning and transfer learning ability for the large-scale pre-trained models. By adopting a generalized low-rank adaptation and re-parameterization framework, GLoRA significantly reduces the number of parameters and computation required for fine-tuning, making it a more resource-efficient and practical method for real-world applications. The experiments conducted on a diverse range of tasks and datasets have substantiated the superiority of GLoRA over existing PEFT techniques, showcasing its scalability and adaptability. Moreover, the ablation studies have provided valuable insights into the inner workings and the relative importance of different GLoRA components. This work not only contributes to the improvement of the fine-tuning process for large-scale pre-trained models but also opens up new avenues for future work, including further exploration of generalized low-rank adaptation techniques, the development of hybrid approaches, and the refinement of search and optimization algorithms. These areas of research may continue to expand the accessibility and efficiency of transfer learning across a broader range of applications.

**Reproducibility**. We provide detailed training recipes in Section 3 and Appendix A. Code is also provided for reproducibility.

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

APPENDIX

## A  HYPERPARAMETERS

Table 6: Learning rate of dataset-specific supernet training on VTAB-1K datastet.

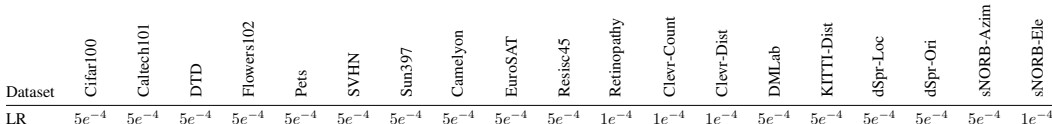

| Dataset | Cifar100 | Caltech101 | DTD | Flowers102 | Pets | SVHN | Sun397 | Camelyon | EuroSAT | Resisc45 | Retinopathy | Clevr-Count | Clevr-Dist | DMLab | KITTI-Dist | dSpr-Loc | dSpr-Ori | sNORB-Azim | sNORB-Ele |
|---|---|---|---|---|---|---|---|---|---|---|---|---|---|---|---|---|---|---|---|
| LR | $5e^{-4}$ | $5e^{-4}$ | $5e^{-4}$ | $5e^{-4}$ | $5e^{-4}$ | $5e^{-4}$ | $5e^{-4}$ | $5e^{-4}$ | $5e^{-4}$ | $5e^{-4}$ | $1e^{-4}$ | $1e^{-4}$ | $1e^{-4}$ | $5e^{-4}$ | $5e^{-4}$ | $5e^{-4}$ | $5e^{-4}$ | $5e^{-4}$ | $1e^{-4}$ |

Our approach necessitate minimal adjustments to hyperparameters, with optimizer hyperparameters being the sole exception, thanks to the inherent search mechanism. Following prior studies (Dehghani et al., 2023; Chen et al., 2022; Zhang et al., 2022), we employ the AdamW optimizer (Loshchilov & Hutter) for all our experiments.

For the hyperparameter search in vision tasks, we primarily concentrate on the exploration of the learning rate for supernet training, limiting our search scope to two potential alternatives: $1e^{-4}$ and $5e^{-4}$. For a detailed account of dataset-specific learning rates, please refer to Table 6. All other training particulars strictly adhere to the exact training policy delineated in the works of (Jie & Deng, 2022; Luo et al., 2023). In the case of few-shot learning datasets and ImageNet, we use learning rates of $5e^{-4}$ and $1e^{-4}$ respectively, as the few-shot learning datasets are smaller as compared to 16-shot ImageNet.

For language modeling experiments we use a learning rate of $2e^{-5}$ with cosine annealing and an equivalent batch size of 32 (using gradient accumulation) for both LoRA and GLoRA. Consequently, LoRA is trained for 3 epochs, and due to the supernet structure of GLoRA, we train it for 15 epochs. This is in line with vision experiments where LoRA is trained for 100 epochs and GLoRA supernet for 500 epochs. We justify these extra training epochs due to the fact that LoRA requires hyperparameter tuning (dropout rate, adaptation layer choice, alpha, etc.) while GLoRA being a searched-based method requires no such tuning. We provide more such method-specific hyperparameters in Appendix D which justifies GLoRA's extra training time.

## B  EVOLUTIONARY SEARCH

Evolutionary search consists of reproduction, crossover, and mutation stages. In our scenario, first, a population of support tensor strategies is embedded in vectors and initialized randomly. Each individual strategy consists of a description of a single subnet. After supernet training, we start to evaluate each individual subnet to obtain its accuracy on the validation set. Among these evaluated subnets we select the top $K$ as parents to produce posterity subnets. The next generation subnets are made by mutation and crossover stages. By repeating this process in iterations, we can find the best parameter-efficient fine-tuned subnet with the best validation performance. We first randomly sample 50 subnets from the supernet and then perform an evolutionary search for 20 and 5 epochs on vision and language tasks, respectively. Each step of random pick / crossover / mutation produces 50 new subnets. The probability for crossover and mutation is set to 0.2. Note that we did not perform any hyperparameter search over the evolution hyperparameters, and hence the performance might even improve after tuning the evolution hyperparameters.

## C  HIERARCHICAL TRANSFORMER

We show the performance of GLoRA on the Swin-B backbone in Table 7. We follow a dataset-specific learning rate search similar to ViT-B and also add GLoRA to the reduction linear layer in Swin architecture to maintain uniformity and avoid architecture-specific tuning. GLoRA can adapt to any layer irrespective of architecture configuration and perform well across tasks and datasets which can be clearly seen in Table 7 where GLoRA outperforms all existing works by a fair margin.

Table 7: Performance on VTAB-1K benchmark with Swin-B pre-trained on ImageNet-21K as the backbone.

| Method | Natural | Specialized | Structured | Average |
|---|---|---|---|---|
| Full | 79.2 | 86.2 | 59.7 | 75.0 |
| Linear | 73.5 | 80.8 | 33.5 | 62.6 |
| BitFit | 74.2 | 80.1 | 42.4 | 65.6 |
| VPT | 76.8 | 84.5 | 53.4 | 71.6 |
| FacT | 82.7 | 87.5 | 62.0 | 77.4 |
| RepAdapter | 83.1 | 86.9 | 62.1 | 77.4 |
| GLoRA | 83.7 | 88.7 | 61.9 | 78.1 |

Table 8: Manual design choices in existing works

| Method | Design Choices/Hyperparameters |
|---|---|
| VPT | Prompt Length, Prompt Location, Prompt Depth |
| AdaptFormer | Adapter Location, Scaling Factor, Hidden dimension, Insertion Form |
| NOAH | VPT choices, Adapter choices, LoRA rank |
| RepAdapter | Adapter Location, Number of groups, Hidden dimension, Adapter variants |
| FacT | Decomposition method, Scaling factor, Decomposition Rank |
| GLoRA | LoRA ranks in search space |

## D    TRAINING TIME

GLoRA, being a search-based approach for PEFT, naturally incurs increased training time due to the requirements of supernet training and evolutionary search. It is, however, critical to underscore that all current methods necessitate a manual search for design choices, as evidenced in Table 8. This necessity significantly inflates the total training time for a specific dataset, due to the broad search within these design choices. GLoRA streamlines this process through an automated evolutionary search mechanism, thus leveraging the benefit of an expansive search space.

Quantitatively GLoRA requires an additional 5.6 folds of training time compared to a single run of LoRA amounting to a total of 142 minutes for each VTAB-1k task. The GPU memory consumption of GLoRA is 13 GB compared to 9 GB for LoRA. Most of it is primarily because GLoRA requires roughly 5 times more epochs than LoRA for appropriate convergence and the additional time is spent on the evolutionary search process. This extra time of GLoRA leads to an average increase of 4.5 % accuracy across 19 vision tasks as compared to LoRA.

## E    SEARCH SPACE

In this section, we undertake the computation of the possible number of subnets within our GLoRA-adapted supernet. Each layer offers $4, 4, 3, 3,$ and $3$ options for the support tensor $\mathbf{A}$, $\mathbf{B}$, $\mathbf{C}$, $\mathbf{D}$, and $\mathbf{E}$, respectively. This results in $432$ possible configurations for a single linear layer. In our implementation, we incorporate $48$ such layers within ViT-B, yielding a total of $432 \times 48 = 20,736$ subnets being explored within GLoRA. This figure can escalate if multiple LoRA ranks coexist within the same search space. For instance, we allow ranks 8 and 4 in our largest GLoRA models, leading to $82,944$ distinct subnets. Furthermore, owing to the phenomenon of weight entanglement as per (Chen et al., 2021), comparable performance is maintained across all subnets, even if they are not all explored during the training of the supernet.

**Visualizations of searched fine-tuning strategy for each layer.** Figure 4 visually shows the distribution of trainable parameters across the four types of linear layers embodied in ViT-B. Notably, the projection layer possesses the minimum quantity of trainable parameters spanning across VTAB-1K categories. Generally, the MLP module hosts a substantially higher number of parameters compared to the MHSA. As anticipated, the structured group necessitates a greater number of parameters for adaptation due to a pronounced domain drift relative to ImageNet-1K (Deng et al., 2009b). Figure 3 illustrates the layerwise configuration of the support tensors as searched by the GLoRA algorithm.

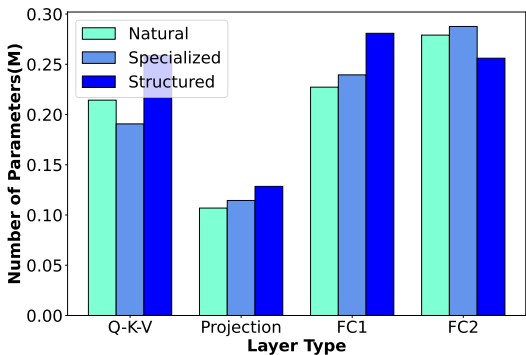

Figure 4: Distribution of GLoRA (0.86M) parameters across layer types on VTAB-1K. Q-K-V and Projection are linear layers in MHSA module and FC1 and FC2 are linear layers in MLP module.

Each support tensor at every layer can potentially undergo 72 distinct adaptations across datasets. Support tensors $\mathbf{D}$ and $\mathbf{E}$ exhibit relatively low adaptation due to the prevalence of `none` adaptations, whereas $\mathbf{A}$ and $\mathbf{B}$ demonstrate a higher number of adaptations, though without a discernible pattern regarding the type of adaptation. It's important to underscore that even a basic scalar can function effectively as a support tensor, enabling GLoRA to maintain superior parameter efficiency despite adapting every linear layer.

## F  SUPPORT TENSOR

In this section, we justify the choices of support tensors in our framework. Consider a linear layer that facilitates the transformation of inputs from a $d_1$ dimensional space to a $d_2$ dimensional space, with a corresponding weight matrix $\mathbf{W}_0 \in \mathbb{R}^{d_2 \times d_1}$. Given that $\mathbf{A}$ is tasked with scaling $\mathbf{W}_0$, $\mathbf{A}$ could feasibly belong to $\mathbb{R}^{d_2 \times d_1}$, $\mathbb{R}^{d_2 \times 1}$, or $\mathbb{R}^{1 \times 1}$. These matrix dimensions are respectively indicative of LoRA, vector, and scalar operations. It's pertinent to note that in scenarios where $\mathbf{A} \in \mathbb{R}^{d_2 \times d_1}$, LoRA is realized via corresponding matrices $\mathbf{A}_d \in \mathbb{R}^{d_2 \times r}$ and $\mathbf{A}_u \in \mathbb{R}^{r \times d_1}$. A parallel scrutiny of other support tensors would result in determining the appropriate support tensor choice, as elaborated in Section 2.3 of the main paper.

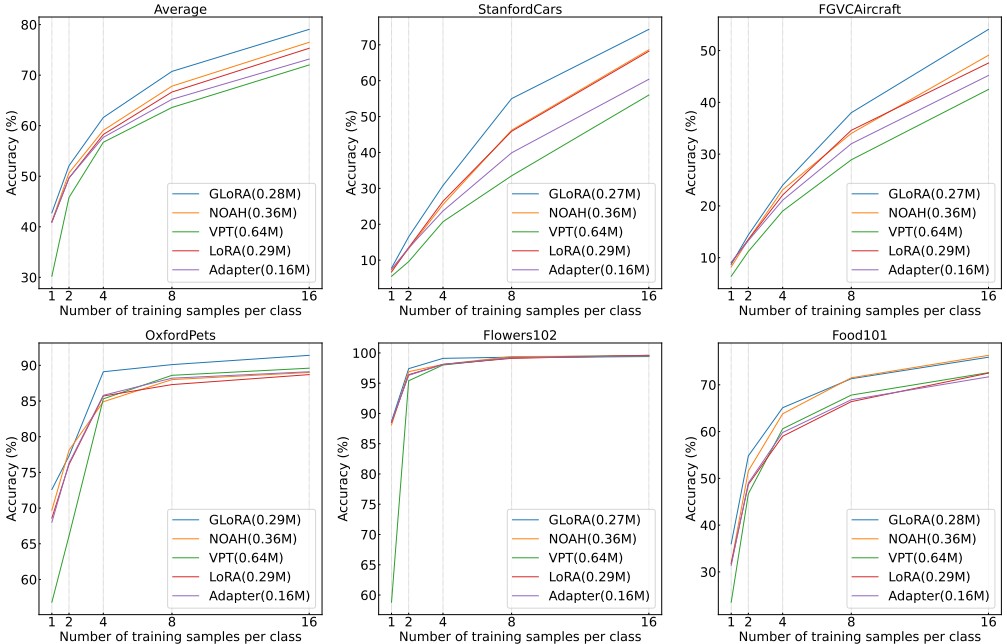

Figure 5: More results on few-shot learning datasets.

# G    MORE RESULTS ON FEW-SHOT LEARNING DATASETS

As shown in 6, the baseline methods include Adapter, LoRA, VPT, NOAH. GLoRA consistently performs better across five datasets and a varying number of training examples per class.

# H    FINE-TUNED EMBEDDING VISUALIZATION

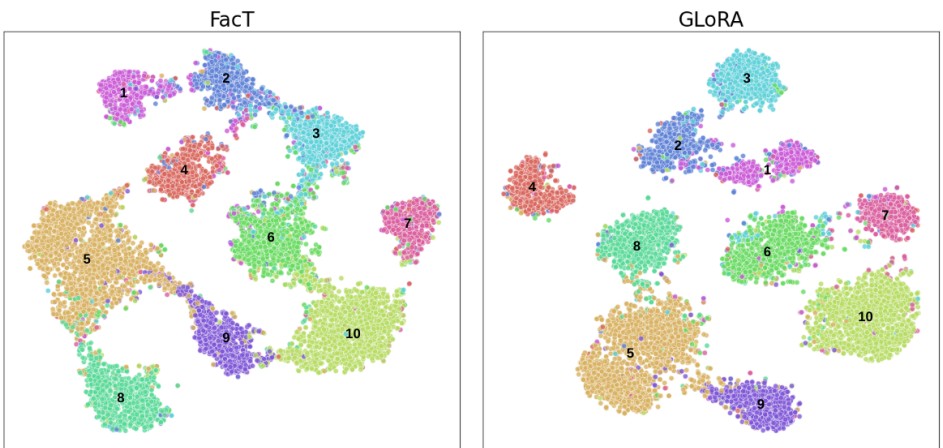

Figure 6: Visualization of features from SVHN dataset by t-SNE (Van der Maaten & Hinton, 2008).

We present feature visualizations of the ViT-B model adapted via GLoRA and FacT (Jie & Deng, 2022) methods applied to the SVHN dataset. We selected FacT as opposed to LoRA (Hu et al.), given that FacT constitutes a direct mathematical enhancement over LoRA and presently represents the state-of-the-art. A clear distinction can be discerned whereby GLoRA exhibits superiorly segregated clusters in comparison to FacT. Further, the delineations are broader, and the clusters demonstrate a higher degree of concentration, signaling the heightened discriminative capacity of the GLoRA-adapted model features.

