# OpenReview forum: "One-for-All: Generalized LoRA for Parameter-Efficient Fine-tuning"
_ICLR.cc/2024/Conference — Submitted to ICLR 2024_

### Official Review · Reviewer_QDBh · 2023-10-28

**Soundness:** 3 good
**Presentation:** 2 fair
**Contribution:** 3 good
**Rating:** 6
**Confidence:** 4

**Summary:**

The paper proposes a general form of low-rank adaptation for vision and language large models, based on a unified formula which the authors claim to encompass serveral previous parameter efficient finetuning methods such as VPT and LoRA. As for training networks with GLoRA, the authors exploit an evolutionary strategy to search for the best subnet after training the supernet. Extensive experiments on vision and language benchmarks show the effectiveness of the propose method.

**Strengths:**

1. The paper rethinks serveral previous PEFT methods and unifies them with a general form, contributing a novel perspective.
2. To obtain the task-specific GLoRA network, the authors first train the supernet and then search for the best subnet.
3. Extensive experiments are conducted on both vision and language benchmarks, and also in few-shot learning and domain generalization, showing the effectiveness GLoRA.

**Weaknesses:**

1. The presentation of the paper could be improved, especially when comparing with previous PEFT methods. The authors could draw figures or list tables to show how existing methods can be integrated into GLoRA framework, e.g. what are the specifications of the A/B/C/D/E support tensors in Eq. (10).
2. I wonder how much training time (supernet training and subnet searching) does GLoRA cost, such that to compare with existing methods more clearly from the perspective of training efficiency.

**Questions:**

1. There exist some typos: 1) "PETL"(maybe PEFT?) at the end of page 3 (first line of Sec. Limitations); 2) 4th line of page 4: "wieght" -> weight.
2. How about the performance if we do not add the weight/bias scaling term: W_0 x A and D x b_0 in Eq. (10) ? Or else, which of the five tensors are really necessary in terms of efficiency and efficacy ?

---

> ### Author Response · Authors · 2023-11-19
> **Response to Reviewer QDBh (Part 1)**
>
> We thank the reviewer for ascertaining the unified and general form of GLoRA which contributes to a novel perspective of existing PEFT methods.
>
> **Q1. The presentation of the paper could be improved, especially when comparing with previous PEFT methods. The authors could draw figures or list tables to show how existing methods can be integrated into GLoRA framework, e.g. what are the specifications of the A/B/C/D/E support tensors in Eq. (10).**
>
> A1. We thank the reviewer for the suggestion and we have worked on improving the overall presentation of the paper. Here we present a table which shows how existing methods can be integrated into GLoRA framework. We have also added this table to the main paper.. We set different support tensors to choices presented in the search space (Eq. (11)) to approximately mimic the behavior of existing methods.
>
> |  Method     |   A    |   B   |   C   |   D   |   E   |
> |--------|:-------:|:-------:|:-------:|:-------:|:--------------:|
> | LoRA | LoRA   | None  | None  | None  | None  |
> | VPT | None   | None  | Vector| None  | None  |
> | SSF | Vector | None  | Vector| Vector| None  |
> |RepAdapter | LoRA   | None  | None  | Vector| None  |
>
> **Q2. I wonder how much training time (supernet training and subnet searching) does GLoRA cost, such that to compare with existing methods more clearly from the perspective of training efficiency.**
>
> A2. When compared to other adaptation methods, the benefits gained with GLoRA in terms of performance improvement as well as generalization across datasets significantly outweigh the additional time spent in the adaptation process.
>
> We demonstrate the superiority of GLoRA by analyzing the performance improvement per unit training time of GLoRA vs. other counterparts. Quantitatively, GLoRA requires an additional 5.6 folds of training time compared to a single run of LoRA amounting to a total of ~142 minutes for each VTAB-1k task. The GPU memory consumption of GLoRA is 13 GB compared to 9 GB for LoRA. Most of it is primarily because GLoRA requires roughly 5 times more epochs than LoRA for appropriate convergence and the additional time is spent on the evolutionary search process. This extra time of GLoRA leads to an average increase of 4.5 % accuracy across 19 vision tasks as compared to LoRA. Beyond this, the biggest benefit comes in terms of generalization across datasets. Additionally, we consider 5 best-performing methods on the VTAB-1k dataset - LoRA, NOAH, FacT, SSF and RepAdapter; and pick the task-specific best-performing models across them. We denote the training time of LoRA as x and the corresponding training time of NoAH, FacT, SSF and RepAdapter are 6x, x, 1.2x and 0.9x respectively. GLoRA's total training time including architecture search is 6.6x. The table below shows the performance and training time of Best-5 methods and GLoRA. This is to indicate that the combined training time of 5 best existing methods is more than that of GLoRA and yet it delivers a superior performance over its counterpart. It is important to note that this gain is reported assuming that the other adaptation methods do not require any hyperparameter search. However, unlike GLoRA, needing minimal hyperparameter search (see Appendix C), some of the other adaptation methods require thorough data-specific hyperparameter search for optimal performance. If we consider this aspect, then the training time required by GLoRA would be significantly lower in a relative sense. We have mentioned the actual training time and memory of GLoRA in the appendix of the main paper.
> | Method   | Training Time | Inference Time | Natural | Specialized | Structured | Average |
> |----------|:---------:|:-------------:|:------------:|:---------:|:-------:|:-------:|
> | NOAH | 6x | &#8593; | 80.3 | 84.9 | 61.3 | 75.5 |
> | Best-5 | 10.1x | &#8593; | 82.5    | 86.7        | 63.3       | 77.5    |
> | GLoRA | 6.6x | - | 83.6    | 87.0        | 63.3       | **78.0**    |

---

> ### Author Response · Authors · 2023-11-19
> **Response to Reviewer QDBh (Part 2)**
>
> **Q3. There exist some typos: 1) "PETL"(maybe PEFT?) at the end of page 3 (first line of Sec. Limitations); 2) 4th line of page 4: "wieght" -> weight.**
>
> A3. We thank the reviewer for pointing out these typos and we have updated the draft accordingly.
>
> **Q4. How about the performance if we do not add the weight/bias scaling term: W_0 x A and D x b_0 in Eq. (10) ? Or else, which of the five tensors are really necessary in terms of efficiency and efficacy ?**
>
> A4. In the construction of GLoRA architectures, each support tensor plays an important role and is necessary to achieve the optimal performance. Among the various supports, A is utilized to scale the weight. B has the role to scale the input and shift the weight. C is the layer-wise prompt serving a similar function of VPT-Deep, D and E are used to scale and shift the bias, respectively. In the table below, we show the performance on CIFAR dataset by setting some support tensors to None, and it is clear that the performance dip is observed for all cases. Clearly, omitting any of these tensors gives sub-optimal performance, implying that each of these is important.
>
> | Settings | Remove A and D | Remove C | Keep All |
> |--------------|:---------:|:------:|:------:|
> | **Accuracy** | 76.0    | 75.9 | 76.5 |
>
> In terms of inference efficiency, GLoRA enjoys zero computational overhead due to the complete structural re-parameterization framework (as shown in Table 4). In terms of training efficiency, it is governed by the set of supports that form the GLoRA architecture. Among LoRA, Vector, Scalar and None attributes, LoRA is most computationally most expensive due to matrix multiplication involved in it, Vector being the next due to dot product operations and Scalar is the most efficient due to simple scalar multiplication. Additionally, since D and E cannot be LoRA, these are the most efficient support tensors. Further, C is the least efficient one due to the absence of scalar multiplication. We have also added these details in the updated draft.

---

> ### Comment · Reviewer_QDBh · 2023-11-21
>
> Thank the authors for their rebuttal, which has resolved my concerns. I would like to keep my original rating.

---

> > ### Author Response · Authors · 2023-11-23
> > **Thank you for your response**
> >
> > Thank you for your response. Please feel free to let us know if you have further concerns.

---

### Official Review · Reviewer_8AV4 · 2023-10-31

**Soundness:** 2 fair
**Presentation:** 3 good
**Contribution:** 2 fair
**Rating:** 5
**Confidence:** 4

**Summary:**

The paper presents Generalized LoRA (GLoRA), an efficient framework for fine-tuning machine learning models. Building on Low-Rank Adaptation (LoRA), GLoRA introduces an advanced prompt module that not only refines pre-trained model weights but also modulates intermediate activations. Uniquely, this prompt module operates individually across each model layer, ensuring versatility across various tasks and datasets.

GLoRA employs a cohesive mathematical strategy to adapt to new tasks, modifying both weights and activations. This methodology positions it strongly for transfer learning, few-shot learning, and domain generalization.

The authors substantiate GLoRA's efficacy through experiments on diverse datasets and tasks, encompassing downstream fine-tuning, few-shot learning, domain generalization, and recent popular LLMs. The results demonstrate GLoRA's superior performance over prior techniques in these areas. Remarkably, despite its heightened capability, GLoRA demands fewer parameters and computations without incurring additional inference costs, akin to LoRA, making it an optimal choice for resource-constrained applications.

**Strengths:**

GLoRA effectively consolidates previous parameter-efficient fine-tuning methods within Equation 10. Importantly, all adjustable support tensors are linear, which makes structural re-parameterization readily accessible.

The paper highlights GLoRA's commendable capacity to generalize across diverse tasks, an invaluable quality in machine learning and a frequently challenging facet of model development.

**Weaknesses:**

Structural re-parameterization requires storing the full set of weights (including bias) for every individual downstream task. This means that as the number of these tasks increases, the storage needs can become prohibitively large. Although this approach might improve inference performance, the substantial storage overhead can be a major impediment for real-world deployment, especially when multiple tuned-models are needed to handle different downstream tasks.

The clarity of the paper is occasionally compromised by abrupt topic transitions, such as the unexpected introduction of "GLoRA with Higher Capacity" in section 2.6, without prior elucidation of terms like H_i and H_ini. A more coherent and gradual introduction of these concepts would enhance readability.

The authors touch on memory and training time costs but fail to provide concrete figures to substantiate their claims. Offering detailed, quantitative data on these costs would provide readers with a clearer picture of GLoRA's practical ramifications.

**Questions:**

Please refer to 'weakness' part.

---

> ### Author Response · Authors · 2023-11-19
> **Response to Reviewer 8AV4 (Part 1)**
>
> We thank the reviewer for poinitng out GLoRA's commendable capacity to generalize across diverse tasks and the cohesive mathematical formulation to adapt to new tasks.
>
> **Q1. Structural re-parameterization requires storing the full set of weights (including bias) for every individual downstream task. This means that as the number of these tasks increases, the storage needs can become prohibitively large. Although this approach might improve inference performance, the substantial storage overhead can be a major impediment for real-world deployment, especially when multiple tuned-models are needed to handle different downstream tasks.**
>
> A1. We implement structural re-parameterization by storing a single base network and multiple GLoRA support tensor weights for each downstream task. Structural re-parameterization of GLoRA takes negligible time (<5s across 100 trials on Llama-7B) and hence depending upon the downstream task, the appropriate support tensors can be re-parameterized in the base network almost instantly during runtime. This saves the storage requirement as GLoRA support tensors require <1% storage as compared to base model and gives a great deployment/inference time saving as shown in Table 4.
>
> **Q2. The clarity of the paper is occasionally compromised by abrupt topic transitions, such as the unexpected introduction of "GLoRA with Higher Capacity" in section 2.6, without prior elucidation of terms like H_i and H_ini. A more coherent and gradual introduction of these concepts would enhance readability.**
>
> A2. We have revised our draft to make the topic transitions smoother. Further, we describe a brief elucidation of terms like H_i and H_uni. The Vapnik-Chervonenkis Dimension (VC Dimension) is a measure of the capacity (complexity, expressiveness) of a set of functions that can be learned by a statistical classification algorithm. It is defined as the cardinality of the largest set of points that the algorithm can shatter. By estimating the VC Dimension of a deep model, we can get an idea of how capable the model is of fitting complex datasets. A very high VC Dimension could indicate that the model has enough capacity to learn the training data perfectly but might overfit and generalize poorly on new data.
>
> **Definition of VC Dimension.** The VC Dimension of a hypothesis class $\mathcal H$ (a set of functions) is the largest number of points that can be shattered by $\mathcal H$. A set of points is said to be shattered by $\mathcal H$ if, for every possible labeling (binary classification) of these points, there exists a hypothesis in $\mathcal H$ that perfectly classifies the points according to that labeling.
> Mathematically, if we have a set of points $S=\{x_1, x_2, \ldots, x_d\}$, the hypothesis class $\mathcal H$ shatters $S$ if:
> \begin{equation}
>     \forall y \in\{0,1\}^d, \exists h \in \mathcal H: \forall i \in\{1,2, \ldots, d\}, h\left(x_i\right)=y_i
> \end{equation}
>
> The VC Dimension, denoted as $\mathbf d_\mathrm{vc}(\mathcal H)$, is the maximum size of any set $S$ that can be shattered by $\mathcal H$. If $\mathcal H$ can shatter a set of size $d$ but cannot shatter any set of size $d+1$, then $\mathbf d_\mathrm{vc}(\mathcal H)=d$. In the context of GLoRA, $\mathcal{H_i}$ denotes the hypothesis space of a randomly sampled subnet and $\mathcal{H}_\mathrm{uni}$ denotes the hypothesis space of the complete supernet.

---

> ### Author Response · Authors · 2023-11-19
> **Response to Reviewer 8AV4 (Part 2)**
>
> **Q3. The authors touch on memory and training time costs but fail to provide concrete figures to substantiate their claims. Offering detailed, quantitative data on these costs would provide readers with a clearer picture of GLoRA's practical ramifications.**
>
> A3. When compared to other adaptation methods, the benefits gained with GLoRA in terms of performance improvement as well as generalization across datasets significantly outweigh the additional time spent in the adaptation process.
>
> We demonstrate the superiority of GLoRA by analyzing the performance improvement per unit training time of GLoRA vs. other counterparts. Quantitatively, GLoRA requires an additional 5.6 folds of training time compared to a single run of LoRA amounting to a total of ~142 minutes for each VTAB-1k task. The GPU memory consumption of GLoRA is 13 GB compared to 9 GB for LoRA. Most of it is primarily because GLoRA requires roughly 5 times more epochs than LoRA for appropriate convergence and the additional time is spent on the evolutionary search process. This extra time of GLoRA leads to an average increase of 4.5 % accuracy across 19 vision tasks as compared to LoRA. Beyond this, the biggest benefit comes in terms of generalization across datasets. Additionally, we consider 5 best-performing methods on the VTAB-1k dataset - LoRA, NOAH, FacT, SSF and RepAdapter; and pick the task-specific best-performing models across them. We denote the training time of LoRA as x and the corresponding training time of NoAH, FacT, SSF and RepAdapter are 6x, x, 1.2x and 0.9x respectively. GLoRA's total training time including architecture search is 6.6x. The table below shows the performance and training time of Best-5 methods and GLoRA. This is to indicate that the combined training time of 5 best existing methods is more than that of GLoRA and yet it delivers a superior performance over its counterpart. It is important to note that this gain is reported assuming that the other adaptation methods do not require any hyperparameter search. However, unlike GLoRA, needing minimal hyperparameter search (see Appendix C), some of the other adaptation methods require thorough data-specific hyperparameter search for optimal performance. If we consider this aspect, then the training time required by GLoRA would be significantly lower in a relative sense. We have mentioned the actual training time and memory of GLoRA in the appendix of the main paper.
> | Method   | Training Time | Inference Time | Natural | Specialized | Structured | Average |
> |----------|:---------:|:-------------:|:------------:|:---------:|:-------:|:-------:|
> | NOAH | 6x | &#8593; | 80.3 | 84.9 | 61.3 | 75.5 |
> | Best-5 | 10.1x | &#8593; | 82.5    | 86.7        | 63.3       | 77.5    |
> | GLoRA | 6.6x | - | 83.6    | 87.0        | 63.3       | **78.0**    |

---

> > ### Comment · Reviewer_8AV4 · 2023-11-22
> >
> > Thank the authors for their reply. Considering the nature of the conference and the novelty of the paper, I believe this is a borderline paper. I decided to keep my original score. However, accepting it would not be a bad choice either.

---

> > > ### Author Response · Authors · 2023-11-23
> > > **Thank you for your response**
> > >
> > > We thank the reviewer for reading our rebuttal and providing a kind response, we are also encouraged by your acknowledgment that our paper is acceptable. While we hold your perspective in high regard, our proposed GLoRA incorporates a new framework in which one formulation can be transformed into various shapes of PEFT paradigms, to unify multiple PEFT methods in a single framework, and we demonstrate its effectiveness over state-of-the-art counterparts. We believe our work carries both novelty (e.g., also mentioned by Reviewer QDBh) and significance. Please feel free to let us know if you have further concerns. We are glad to address any of them.

---

### Official Review · Reviewer_1LBP · 2023-11-03

**Soundness:** 3 good
**Presentation:** 3 good
**Contribution:** 3 good
**Rating:** 6
**Confidence:** 4

**Summary:**

This paper presents a new PEFT module named Generalized LoRA (GLoRA), which can be applied to different tasks. The authors claim that GLoRA is more general than existing PEFT modules since it can facilitate efficient parameter adaptation by employing a more scalable structure search. Moreover, the authors conduct multiple experiments including few-shot learning and domain generalization tasks to demonstrate the effectiveness of GLoRA.

**Strengths:**

1. GLoRA has a re-parameterization design. It is more similar to LORA than Adapter. It makes GLoRA more flexible since it does not need to change the structure of the original backbone. And it incurs no extra inference cost.
2. GLoRA integrates multiple methods and can perform similar effects as most of the existing PEFT modules.
3. The authors conduct multiple experiments to demonstrate the generality and effectiveness of GLoRA.

**Weaknesses:**

1. It seems that GLoRA is not general, since it has an evolutionary search procedure to obtain the suitable components. The idea is similar to Neural Prompt Search [1]. GLoRA is not a fixed design as existing modules, which might limit its practicality.
2. GLoRA has a large search space, which might yield huge time costs. However, the authors have not mentioned the actual training time and memory cost of GLoRA, which is very important for PEFT modules.
3. The authors introduce multiple PEFT modules including AdaptFormer, LoRA, etc. So could GLoRA simulate all of these modules? As far as I know, these modules are applied on different layers (LoRA in multi-head self-attention layers, while AdaptFormer in MLP layers). And which layer is GLoRA applied in practice?

[1] Neural Prompt Search. In https://arxiv.org/abs/2206.04673.

**Questions:**

See "Weaknesses".

---

> ### Author Response · Authors · 2023-11-19
> **Response to Reviewer 1LBP (Part 1)**
>
> We thank the reviewer for finding our work to be more flexible than existing works owing to the structural re-parameterizable framework. We would also like to appreciate the reviewer for pointing out the thorough experiments conducted showing the generality and effectiveness of GLoRA.
>
> **Q1. It seems that GLoRA is not general, since it has an evolutionary search procedure to obtain the suitable components. The idea is similar to Neural Prompt Search. GLoRA is not a fixed design as existing modules, which might limit its practicality.**
>
> A1. We understand the concern of the reviewer, however, our use of 'general' for GLoRA does not refer to one set of adapters that can be plugged on top of a model. As we have shown, the performance of such a method varies significantly across datasets. Rather, GLoRA is general in the sense that the combination of supernet training and subnet search which form an inherent part of the method, result in well performing data-specific architectures. Although GLoRA is a search based method similar to [1], the core novelty of GLoRA lies in the unified re-parameterizable formulation as presented in Eq. 10. [1] attempts to search for VPT, LoRA and Adapter in a single framework, by defining them distinctly in the supernet whereas GLoRA doesn't define any existing adapter/module in the network explicitly but proposes a much 'general formulation' which implicitly mimics the behavior of many existing works.
>
> Regarding the limitation in terms of practicality, GLoRA identifies the right combination of support tensors/modules/prompts for each dataset. Any optimized network obtained from GLoRA can be absorbed into the base architecture using structural re-parameterization rendering superior or the same practical throughput as existing works. Thus, we do not see any limitations of the approach compared to the other baseline methods such as LoRA, SSF, etc.
>
> **Q2. GLoRA has a large search space, which might yield huge time costs. However, the authors have not mentioned the actual training time and memory cost of GLoRA, which is very important for PEFT modules.**
>
> A2. As highlighted by the reviewer, the large search space of GLoRA indeed increases the associated training time compared to the other adaptation methods. However, it is important to note that the overall time spent on the adaptation of models using GLoRA is still substantially very small when compared to fine-tuning the full model. Further, when compared to other adaptation methods, the benefits gained with GLoRA in terms of performance improvement as well as generalization across datasets significantly outweigh the additional time spent in the adaptation process.
>
> We demonstrate the superiority of GLoRA by analyzing the performance improvement per unit training time of GLoRA vs. other counterparts. Quantitatively, GLoRA requires an additional 5.6 folds of training time compared to a single run of LoRA amounting to a total of ~142 minutes for each VTAB-1k task. The GPU memory consumption of GLoRA is 13 GB compared to 9 GB for LoRA. Most of it is primarily because GLoRA requires roughly 5 times more epochs than LoRA for appropriate convergence and the additional time is spent on the evolutionary search process. This extra time of GLoRA leads to an average increase of 4.5 % accuracy across 19 vision tasks as compared to LoRA. Beyond this, the biggest benefit comes in terms of generalization across datasets. Additionally, we consider 5 best-performing methods on the VTAB-1k dataset - LoRA, NOAH, FacT, SSF and RepAdapter; and pick the task-specific best-performing models across them. We denote the training time of LoRA as x and the corresponding training time of NoAH, FacT, SSF and RepAdapter are 6x, x, 1.2x and 0.9x respectively. GLoRA's total training time including architecture search is 6.6x. The table below shows the performance and training time of Best-5 methods and GLoRA. This is to indicate that the combined training time of 5 best existing methods is more than that of GLoRA and yet it delivers a superior performance over its counterpart. It is important to note that this gain is reported assuming that the other adaptation methods do not require any hyperparameter search. However, unlike GLoRA, needing minimal hyperparameter search (see Appendix C), some of the other adaptation methods require thorough data-specific hyperparameter search for optimal performance. If we consider this aspect, then the training time required by GLoRA would be significantly lower in a relative sense. We have mentioned the actual training time and memory of GLoRA in the appendix of the main paper.
> | Method   | Training Time | Inference Time | Natural | Specialized | Structured | Average |
> |----------|:---------:|:-------------:|:------------:|:---------:|:-------:|:-------:|
> | NOAH | 6x | &#8593; | 80.3 | 84.9 | 61.3 | 75.5 |
> | Best-5 | 10.1x | &#8593; | 82.5    | 86.7  | 63.3   | 77.5    |
> | GLoRA | 6.6x | - | 83.6    | 87.0| 63.3  | **78.0**    |

---

> ### Author Response · Authors · 2023-11-19
> **Response to Reviewer 1LBP (Part 2)**
>
> **Q3. The authors introduce multiple PEFT modules including AdaptFormer, LoRA, etc. So could GLoRA simulate all of these modules? As far as I know, these modules are applied on different layers (LoRA in multi-head self-attention layers, while AdaptFormer in MLP layers). And which layer is GLoRA applied in practice?**
>
> A3. Indeed GLoRA is capable of simulating multiple existing PEFT modules which include SSF, VPT, LoRA and RepAdapter. For example, in equation 10, setting all support tensors but A to None and setting A as LoRA would inherently give the classic LoRA model. To mimic SSF, A, D and E can be set to vector and other all support tensors to none. Similarly, C simulates the learning mechanism of a layerwise trainable prompt similar to VPT. Note that GLoRA cannot simulate AdaptFormer since it has a non-linearity in the parallel block inhibiting structural re-parameterization which is a core aspect of GLoRA. In the table below, we show how GLoRA is able to approximately mimic the behavior of many existing works by setting support tensors to specific attributes -
>
> |  Method     |   A    |   B   |   C   |   D   |   E   |
> |--------|:-------:|:-------:|:-------:|:-------:|:--------------:|
> | LoRA | LoRA   | None  | None  | None  | None  |
> | VPT | None   | None  | Vector| None  | None  |
> | SSF | Vector | None  | Vector| Vector| None  |
> |RepAdapter | LoRA   | None  | None  | Vector| None  |
>
> For vision tasks GLoRA is applied to all linear layers (MLP as well as the attention modules). In case of NLP tasks, for a fair comparison to LoRA, we apply GLoRA only in the multi-head self-attention layers. These details are mentioned in Section 3.2 of the main paper. However, in general, GLoRA can also be applied to the MLP layers, and we expect that similar to LoRA, it will improve the performance further [2].
>
> [1] Neural Prompt Search.
>
> [2] QLoRA: Efficient Finetuning of Quantized LLMs

---

> > ### Comment · Reviewer_1LBP · 2023-11-23
> >
> > Thank you for your response, which has addressed my concern. I would like to raise my score to 6.
> >
> > If compared to LoRA, the training time of GLoRA might be a main shortage (6.6x), and the GPU memory consumption is also higher. Nevertheless, the performance gains are significant, especially when compared to the search method NOAH. And the cost is close to NOAH.
> >
> > I agree with other reviewers that this is a borderline paper. However, I think the topic of designing more general PEFT modules is interesting and can inspire future research.

---

> ### Author Response · Authors · 2023-11-23
> **Thank reviewer for your encouraging comments**
>
> Thank you for your kind response and encouraging comments. To provide further context, our approach only searches adaptor components but will not search for the optimal set of hyperparameters, such as rank, dropout, or alpha as in LoRA, enabling an automatic adaptation. This indicates that there is further potential to improve the performance of our framework with better configurations. We would like to address any further concerns that you might have and continue to solidify this paper.

---

### Official Review · Reviewer_jzkZ · 2023-11-06

**Soundness:** 4 excellent
**Presentation:** 4 excellent
**Contribution:** 3 good
**Rating:** 5
**Confidence:** 5

**Summary:**

The paper discusses an enhancement to Low-Rank Adaptation (LoRA), which is call GLoR and can be a flexible approach to optimize model inference results. Experiments on llama and vit shows that GLoRA can improve over original LoRA consistently.

**Strengths:**

This paper is clearly presented and well organized. The authors also provide a detailed discussion of related works and variants.

GLoRA referneces the inspirations from RepVGG, which introduces fusable parameters during training to improve model capacity. This can generally bring improvements without extra inference cost as shown in the experiments.

GLoRA offers a unifed framework that includes multiple fine-tuning paradigms and provides a more generalized prompt mdule design per layer. The final scheme is searched via evolutional algorithms, providng better capacity and flexibility.

**Weaknesses:**

The authors claim that GLoRA can be "seamlessly integrate into the base network", but it seems such design is for linear layer only. But there are many other type of operators like conv / normalization layers. How can GLoRA be combined with those layers?

The evolutional search (Sec 2.4) is crucial for GLoRA as it decides which layer and scheme to use during fine-tuning. However, the details of the search and final chosen paradigms are not clearly discussed in the main paper.

As the abstract emphasiss the llama experiments. The table 2 is not solid neough to support the authors' claim. For example, the mean and variance is not included; the number of learnable paramters is missing; the lora baseline for llama-v2 is not reported. Seems like the experiments are rushed and may not be solid.

**Questions:**

No extra inference cost is the novelty from RepVGG or LoRA, which should be claimed as the contribution of GLoRA

The main experiments are based on ViT but abstract emphasis for llama. Please make the claim conssitent.

---

> ### Author Response · Authors · 2023-11-19
> **Response to Reviewer jzkZ (Part 1)**
>
> We thank the reviewer for finding our work to consistently improve performance over previous method in both vision and language tasks as well as finding GLoRA to be more flexible, have higher capacity and provide an unified framework for PEFT.
>
> **Q1. The authors claim that GLoRA can be "seamlessly integrate into the base network", but it seems such design is for linear layer only. But there are many other type of operators like conv / normalization layers. How can GLoRA be combined with those layers?**
>
> A1. We would like to point out to the reviewer that GLoRA is an enhanced version of LoRA and the simplicity of our approach allows us to use it in a similar fashion as LoRA. We further refer the reviewer to the PEFT Library [1] which implements LoRA for CNN and other layers. Since GLoRA uses a search space as defined in Eq. 11 which comprises LoRA, Vector, Constant and None attributes; the exact PEFT implementation can be used to implement GLoRA as well. In the context of a CNN layer featuring *i* input channels, *o* output channels, and *k* kernels, and for the support tensor *A*, we establish convolutional layer weights denoted as *A$_d$* (*o $\times$ r $\times$ 1 $\times$ 1*) and *A$_u$* (*r $\times$ i $\times$ k $\times$ k*). In instances of LoRA with a specified rank *r$_1$* < *r*, we employ *A$_{dr1}$* (*o $\times$ r$_1$ $\times$ 1 $\times$ 1*) and *A$_{ur1}$* (*r$_1$ $\times$ i $\times$ k $\times$ k*), derived through indexing from *A$_d$* and *A$_u$*, respectively, for consecutive low-rank convolution operations. For a Vector attribute, *A* (*1 $\times$ o $\times$ 1 $\times$ 1*) is indexed from *A$_d$*, facilitating vector multiplication. Similarly, in the case of a constant attribute *A* (*1 $\times$ 1 $\times$ 1 $\times$ 1*), indexing is applied for scalar multiplication with the original weight matrix from *A$_d$* during computation. This approach aligns seamlessly with established conventions in linear layer implementation.
>
> Further, since normalization layers have relatively very less number of parameters, PEFT is commonly not employed for them. For example - BatchNorm2D with *c* channels, has only *2 $\times$ c* (weight and bias) total number of parameters which is negligible as compared to Conv2D which has *o $\times$ i $\times$ k $\times$ k* (weight) + *o* (bias). Thus, unlike the conv layers, these do not require parameter-efficient adaptation.
>
> **Q2. The evolutional search (Sec 2.4) is crucial for GLoRA as it decides which layer and scheme to use during fine-tuning. However, the details of the search and final chosen paradigms are not clearly discussed in the main paper.**
>
> A2. We agree that evolutionary search is important and the corresponding details require discussion. We have described the details of the approach in Appendix B of the paper. In general, we did not observe any uniform structure of the final network across datasets. An average distribution of the network across all the datasets is presented in Figures 3 and 4 of the paper. Upon the recommendation of the reviewer we have moved details of the evolutionary search in the main paper.

---

> > ### Author Response · Authors · 2023-11-19
> > **Response to Reviewer jzkZ (Part 2)**
> >
> > **Q3. As the abstract emphasis the llama experiments. Table 2 is not solid enough to support the authors' claim. For example, the mean and variance is not included; the number of learnable parameters is missing; the lora baseline for llama-v2 is not reported. Seems like the experiments are rushed and may not be solid.**
> >
> > A3. We would like to reassure the reviewer that the language experiments were thoroughly conducted and completed well before a certain time of the submission deadline, ensuring they are comprehensively prepared and presented. We performed three searches with different random seeds on LLaMA-7B using GLoRA on the shareGPT dataset (the results are shown in the table below), and observed a mean and variance of 52.66 and 0.0022, respectively, indicating that GLoRA shows consistent performance with respect to initialization. We have updated the submission to reflect this. In general, mean and variance together serve well the purpose of denoting the performance and stability of a method. In the case of fine-tuning experiments, the weights of the base model are frozen and the only randomness introduced into the learning system is from the additional weights of the added adaptation module of GLoRA. In our observation, the initialization of these weights has a negligible effect on the final results, and to our knowledge, even the previous fine-tuning methods such as LoRA, SSF and AdaptFormer, do not report the mean and variance of the experiments.
> >
> > | ARC (25-s) | HellaSwag (10-s) | MMLU (5-s) | TruthfulQA (0-s) | Average |
> > |:---------:|:---------:|:-------:|:-------------:|:------:|
> > | 53.2 | 77.4 | 36.2 | 43.9 | 52.7 |
> > | 53.1 | 77.6 | 36.8 | 43.1 | 52.6 |
> > | 53.1 | 77.4 | 36.7 | 43.8 | 52.7 |
> >
> > We have also added the details on the number of final subnet parameters for LoRA and GLoRA in the table below for LLaMA2-7B. It is important to note that not all supernet parameters are jointly trained in GLoRA and in each iteration only one subnet is used for gradient propogation. It can also be seen that, GLoRA ends up with lesser number of final subnet parameters as compared to LoRA. Note that during the inference, these get absorbed in the re-parameterized final network and no overhead is observed for either LoRA or GLoRA. We have additionally provide the LoRA baseline for LLaMA-2-7B in the main paper and also updated Table 2. to include the parameter count. We have also provided anonymous links to the LLaMA-1 and LLaMA-2 models fine-tuned using GLoRA and plan on open-sourcing the weights and code. We hope that the extra details make the language model experiments more convincing to the reviewer.
> >
> > | Method  | Final #Param | Supernet #Param | Inference #Param | Average Perf. |
> > |---------|:----------------------:|:-------------------:|:-----------------------:|:--------------:|
> > | LoRA    | 3.1M                 | -             | 0M                    | 49.8         |
> > | GLoRA   | **1.8M**                 | 8.6M              | 0M                    | **52.7**         |
> >
> > LLaMA-1 - [anonymous link](https://drive.google.com/drive/folders/124pePmfTG3HW7n5QXeq2MlsfJ2SuN70E?usp=sharing).
> >
> > LLaMA-2 - [anonymous link](https://drive.google.com/drive/folders/1-3tUPlLiEB2CJnENnViTuGA1fDKkrlx8?usp=sharing).
> >
> > **Q4. No extra inference cost is the novelty from RepVGG or LoRA, which should be claimed as the contribution of GLoRA.**
> >
> > A4. To our understanding, the reviewer is implying that similar to RepVGG and LoRA, GLoRA leads to no additional inference cost. We have discussed this in the last line of Contribution 2 in the Introduction part of the paper. Additionally, Table 4 highlights the inference efficiency of GLoRA with other methods.
> >
> > **Q5. The main experiments are based on ViT but abstract emphasis for llama. Please make the claim consistent.**
> >
> > A5. We appreciate the feedback. The strength of GLoRA lies in its rapid adaptation of both vision and language models, as demonstrated through comprehensive experiments on vision tasks. Consequently, we have modified the abstract in accordance with the reviewer's suggestion in the revised submission.
> >
> > [1] PEFT: State-of-the-art Parameter-Efficient Fine-Tuning methods

---

> > > ### Comment · Reviewer_jzkZ · 2023-11-21
> > >
> > > Thanks for the reponses. After carefully reading the rebuttal, I decide to keep my original rating.

---

> > > > ### Author Response · Authors · 2023-11-23
> > > > **Response to Reviewer jzkZ**
> > > >
> > > > Please feel free to let us know if you have any further concerns. We are glad to address any of them. Thanks again for your reply.

---

### Author Response · Authors · 2023-11-19
**Summary of Revision**

We thank all the reviewers for reading our work and providing many insights like higher flexibility of GLoRA (Reviewer jzkZ, 1LBP), general unified framework (Reviewer QDBh, jzkZ), comprehensive experiments (Reviewer 8AV4, QDBh) among others. We have addressed all the questions posed by the reviewers and also updated our main paper with light blue text denoting restructuring of text and dark blue denoting additional content. Overall, we have addressed the following important aspects in our revision:

- Actual training time, memory cost and overall training efficiency. (Reviewer 1LBP, 8AV4, QDBh)
- Clarity over the simulation of existing methods using GLoRA's formulation. (Reviewer 1LBP, QDBh)
- Overall presentation of the paper inlcuding important evolutionary search details and gradual topic transition. (Reviewer jzkZ, 8AV4)
- Solidness of language modelling experiments. (Reviewer jzkZ)

---

### Meta-Review · Area_Chair_oLwW · 2023-12-15

**Metareview:**

The work introduced an improved method of Low-Rank Adaptation (LoRA), which is called GLoRA. It is a flexible approach to optimize model inference results. Experiments on llama and ViT show that GLoRA can improve over the original LoRA consistently.

The paper rethinks several previous PEFT methods and unifies them with a general form, contributing a novel perspective.
However, there are several concerns about the clarity of the paper presentation, the experimental results (especially more detailed study on the memory and training time costs). We encourage the authors to incorporate all the feedbacks for future submissions.

**Justification For Why Not Higher Score:**

The work does not seem to be ready for publication at the current stage

**Justification For Why Not Lower Score:**

n/a

---

### Decision · Program_Chairs · 2024-01-16

Reject